# Oral D-Aspartate Treatment Improves Sperm Fertility in Both Young and Adult B6N Mice

**DOI:** 10.3390/ani12111350

**Published:** 2022-05-25

**Authors:** Marcello Raspa, Renata Paoletti, Manon Peltier, Mohamed Majjouti, Michele Protti, Laura Mercolini, Esther Mahabir, Ferdinando Scavizzi

**Affiliations:** 1National Research Council (IBBC), CNR-Campus International Development (EMMA-INFRAFRONTIER-IMPC), 00015 Monterotondo Scalo, Italy; marcello.raspa@cnr.it; 2Allevamenti Plaisant SRL, 00128 Rome, Italy; renata.paoletti@emma.cnr.it; 3Comparative Medicine, Center for Molecular Medicine Cologne (CMMC), Faculty of Medicine and University Hospital Cologne, 50931 Cologne, Germany; manon.peltier@uk-koeln.de (M.P.); mohamed.majjouti@uk-koeln.de (M.M.); esther.mahabir-brenner@uni-koeln.de (E.M.); 4Department of Pharmacy and Biotechnology (FaBiT), Alma Mater Studiorum, University of Bologna, 40126 Bologna, Italy; michele.protti2@unibo.it (M.P.); laura.mercolini@unibo.it (L.M.)

**Keywords:** D-Aspartate, spermatozoa, sperm fertility, in vitro fertilization (IVF), 3R

## Abstract

**Simple Summary:**

Investigations concerning the impact of D-Aspartate on fertility suggest that it has a positive influence on the in vitro fertilization rate in young C57BL/6N mice. Here, we demonstrated that adult C57BL/6N mice that received an oral treatment of D-Aspartate also have a higher fertilizing capability and the quality of their spermatozoa increased after only two weeks of treatment. Hence, this study gives us new insights on the role of D-Aspartate in the regulation of the reproductive activity in both young and adult mice.

**Abstract:**

D-Aspartate (D-Asp) treatment improved the fertility of young male C57BL/6N mice in vivo revealing a direct role on capacitation, acrosome reaction, and fertility in vitro in young males only. We investigated whether the positive effect of D-Asp on fertility could be extended to adult males and evaluated the efficacy of a 2- or 4-week-treatment in vivo. Therefore, 20 mM sodium D-Asp was supplied in drinking water to males of different ages so that they were 9 or 16 weeks old at the end of the experiments. After sperm freezing, the in vitro fertilization (IVF) rate, the birth rate, hormone levels (luteinizing hormone (LH), epitestosterone, and testosterone), the sperm quality (morphology, abnormalities, motility, and velocity), the capacitation rate, and the acrosome reaction were investigated. Oral D-Asp treatment improves the fertilizing capability in mice regardless of the age of the animals. Importantly, a short D-Asp treatment of 2 weeks in young males elevates sperm parameters to the levels of untreated adult animals. In vivo, D-Asp treatment highly improves sperm quality but not sperm concentration. Therefore, D-Asp plays a beneficial role in mouse male fertility and may be highly relevant for cryorepositories to improve mouse sperm biobanking.

## 1. Introduction

Murine genetic material can be stored and distributed as cryopreserved sperm [1,2,3,4,5]. A challenge with cryopreserved-thawed sperm is that survival is highly variable even when the same protocol is used [6,7]. Amelioration of cryo-injury has been the concern of numerous studies in the last decades. However, most of them have focused on in vitro strategies using antioxidants or agents that enhance membrane fluidity [8,9]. Recently, our group reported that the addition of D-Aspartate (D-Asp) to cryopreserved-thawed spermatozoa in vitro improves the fertility of young mice by increasing both the sperm capacitation and the acrosome reaction [10].

D-Aspartic acid is an endogenous amino acid that is found in the nervous and the neuroendocrine systems in many species [11,12]. The role of D-Asp on reproductive parameters in male mice has only recently begun to be the focus of investigations. In the mouse testis, D-Asp levels are regulated in a spatiotemporal manner, increasing with growth, and it functions as a modulator of spermatogenesis [13]. In a recent study, our group used a systemic approach based on oral treatment of 7-week-old mice with 20 mM D-Asp [14]. The results showed that D-Asp treatment increased the total sperm motility, the hormone levels (luteinizing hormone (LH), epitestosterone, and testosterone), the sperm quality, as well as the in vitro fertilization (IVF) rates after 2 and 4 weeks when the males were 9 and 11 weeks old, respectively [14]. One interesting finding was that oral administration of D-Asp for 6 weeks did not increase IVF. It was not clear whether this result was due to the duration of treatment or the age of the males at sacrifice.

Usually, males used for cryopreservation are at least 13 weeks old [1]. However, in some cases, it is advisable to accelerate cryopreservation of spermatozoa for health or economic reasons as well as subfertility challenges. In this study, young (9-week-old mice, usually too young for routine sperm cryopreservation) and adult male mice (16-week-old mice, sexually mature) were euthanized after they were treated orally with 20 mM sodium D-Asp for 2 weeks (7-week-old or 14-week-old mice at the beginning of the treatment) or 4 weeks (5-week-old or 12-week-old mice at the beginning of the treatment). Cryopreserved spermatozoa were used to simulate routine work. Furthermore the use of fresh spermatozoa would make multiple analyses on the same sample extremely difficult. Furthermore, the use of cryopreserved spermatozoa is considered the most widespread system for conserving mutant lines in cryorepositories and it is extremely important to improve its efficiency, thereby increasing sperm quality (morphology, abnormalities, motility, and velocity) and their fertilizing capacity. In the present study, the IVF rate, the birth rate, the hormone levels (LH, epitestosterone, and testosterone) as well as the sperm quality were analyzed. In addition, we investigated two maturation processes, the capacitation rate followed by the acrosome reaction, both of which are essential for the spermatozoa to fuse with the oocyte and to penetrate the zona pellucida. We measured the capacitation rate and the acrosome reaction immediately after thawing and after 1 h of incubation, thereby simulating the routine duration for capacitation.

## 2. Materials and Methods

### 2.1. Mice and Husbandry

C57BL/6N (B6N) mice were housed in individually ventilated cages (Tecniplast, Gazzada, Italy) in the European Mouse Mutant Archive (CNR-EMMA)-Infrafrontier facility at the Consiglio Nazionale delle Ricerche (Monterotondo Scalo, Rome, Italy). All animals were housed under controlled lighting conditions (daily light period, 07.00–19.00 h), at 20 ± 2 °C, with *ad libitum* access to feed (4RFN and Emma 23, Mucedola) and chlorinated filtered water. Mice were tested for micro-organisms every 3 months using 6- to 8-week-old B6N sentinels. Serology was performed according to the FELASA recommendations [15].

### 2.2. Treatment of Mice

A 20 mM solution of sodium D-Aspartate was prepared by diluting a 1M sodium D-Asp solution, composed of 13.3 g of D-Aspartic acid (Sigma Aldrich, cat #219096) neutralized with NaOH in 100 mL distilled water, at a final pH of 6.6–7.5 in drinking water. Young male mice were treated at 7 or 5 weeks of age for 2 or 4 weeks, respectively, so that they were 9 weeks old at the end of the study. Adult 14- or 12-week-old males were treated for the same duration so that all were 16 weeks old at the end of the study. In both experiments, drinking water without D-Asp was administered to control mice.

During the experiments, all males were housed singly. For each of the two age groups, 9 or 16 weeks, 3 males were used for timepoint 0 (basal values) and 6 males (3 controls and 3 D-Asp-treated) were used for each timepoint (2 or 4 weeks). Therefore, a total of 30 males were used for sperm cryopreservation (Figure 1).

### 2.3. Sperm Cryopreservation

Sperm cryopreservation was performed using a cryoprotective medium (CPM) [1]. The CPM consisted of 18% *w*/*v* raffinose (Sigma Aldrich, cat #R7630), 3% *w*/*v* skim milk (BD Diagnostics, cat # 232100), and 477 μM monothioglycerol (MTG, Sigma Aldrich, cat #M6145). Spermatozoa collected from the *caudae epididymides* and *vasa deferentia* were allowed to disperse from the tissues for 10 min at 37 °C. After loading into 0.25 mL French straws (IMV Technologies, L’Aigle, France), the straws were sealed, exposed to liquid nitrogen vapors for 10 min, and then stored in LN_2_.

### 2.4. In Vitro Fertilization (IVF)

IVF was executed as reported previously [14]. Briefly, cryopreserved sperm was thawed at 37 °C for 30 s, transferred into 500 μL of IVF medium (Cook’s medium, K-RVFE-50, Cook Medical, Brisbane, Australia), and incubated for 40 min in a 5% CO_2_ incubator at 37 °C (Thermo Electron Corporation, Milano, Italy) to induce capacitation. A final sperm concentration of 2 to 6 × 10^5^ spermatozoa/mL was used. The cumulus oocyte complexes (COCs) were collected from four-week-old B6N females 12 to 14 h after superovulation elicited by intraperitoneal injections of 5IU PMSG (Intervet, Milan, Italy) and 5IU hCG (Intervet) 48 h later.

The COCs of 4 females were divided between control and D-Asp-treated IVF dishes so that each group received oocytes from the same females (Table 1). After 4 h of sperm-oocytes incubation, oocytes were washed through three 200 µL drops of Cook’s medium and incubated overnight. On the following day, the 2-cell embryos were collected. The IVF rate, which is expressed as the percentage of 2-cell embryos in relation to the number of oocytes used, was calculated as the mean of 8 fertilization dishes, as previously described [14]. For each timepoint, 2 and 4 weeks, and for each of the 2 age groups, 32 females (16 for each group) were euthanized. In addition, for each age group, 16 females were used for timepoint 0. In total, 160 females were used for IVF.

### 2.5. Embryo Transfer

The embryo quality was examined reporting the percentage of pups born after embryo transfer. The 2-cell embryos generated by IVF were surgically transferred into the oviduct of Day 0.5 pseudopregnant Crl:CD1(ICR) female mice, as described [10,14]. For each group, 4 recipients were used, each receiving 15 embryos. The birth rate is the percentage of pups born compared to the number of embryos transferred. For each of the 2 age groups and 2 timepoints (2 and 4 weeks), embryo transfer was performed with 8 recipients (4 controls and 4 D-Asp treated). A total of 32 recipients were used for embryo transfer.

### 2.6. Measurement of Hormones

Hormone analysis was performed with the same males that were used for sperm cryopreservation (i.e., 3 control males and 3 D-Asp-treated males were used at each timepoint in both age groups) with an exception at timepoint 0, at which only 2 males were used for the 9-week-old group (Figure 1). To reduce circadian variation, all animals were euthanized and all samples were collected between 8.00 a.m. and 9.00 a.m. After exposure of mice at 0, 2, or 4 weeks of treatment to carbon dioxide and via intracardial puncture, blood from control or D-Asp-treated males was collected and pooled according to timepoint, treatment, and age. Then, the testes were removed and analyzed individually. Testes and serum samples were collected and stored at −80 °C until the hormones LH, epitestosterone, and testosterone were analyzed, as previously described [14]. An enzyme-linked immunosorbent assay (ELISA) was used to assess LH levels in testes and serum and liquid chromatography coupled to tandem mass spectrometry (LC-MS/MS) was used to assay epitestosterone and testosterone.

### 2.7. Assessment of Sperm Parameters

Sperm parameters were measured with aliquots of the same sperm samples that were used previously for the IVF work. Two straws were used for sperm motility measurements and morphological analyses. Another straw was used for the determination of the capacitation rate and the acrosome reaction.

#### 2.7.1. Measurement of Sperm Concentration and Motility

As described above, spermatozoa were thawed and released into a pre-warmed sampling tube (Carl Roth GmbH & Co. KG, Karlsruhe, Germany). At 37 °C in a moisture-saturated atmosphere of 5% CO_2_ and 95% air in an incubator (Sanyo-Biomedical Ewald Innovationstechnik GmbH, Bad Nenndorf, Germany), they dispersed for 55 min. By gentle pipetting, a homogeneous sperm suspension was made. Afterwards, 2 µL of the sperm suspension was diluted 1:50 with Cook’s medium. The sperm concentration and motility patterns were both analyzed in duplicate with the Hamilton Thorne IVOS computerized semen analyzer (Hamilton Thorne, Beverly, MA, USA) operating at 30 video frames per sec (60 Hz). For each sample analyzed, a total of 10 fields were recorded. Immotile spermatozoa had an average velocity of <7.4 µm/s. The kinematic data recorded included the overall percentage of motile spermatozoa, which moved at a velocity of >7.4 μm/s in any direction, the percentage of progressive motile spermatozoa having a path velocity >50 μm/s, and a straightness ratio >80%. In addition, the velocity of the spermatozoa was categorized into rapid (velocity >50 µm/s), medium (7.4 µm/s < velocity ≤ 50 µm/s), slow (0 µm/s < velocity ≤ 7.4 µm/s), and static (0 µm/s).

#### 2.7.2. Measurement of Sperm Morphology

Spermatozoa were smeared onto a glass slide, they were then air-dried and examined at a magnification of 40X under bright-field microscopy. The morphology of the spermatozoa was evaluated as described previously [16,17,18]. Spermatozoa were considered normal when they did not present any morphological defects (normal head, straight tail, and no cytoplasmic drop) and abnormal when they showed at least one defect of the head (amorphous, detached, or ectopic attachment of the flagella), of the tail (absent, proximal or distal bent of the midpiece, or bifurcated), and/or the presence of cytoplasmic drops (heavy or light-type). For each timepoint, we counted a total of 200 spermatozoa in duplicate and determined the percentage of abnormal spermatozoa.

#### 2.7.3. Determination of the Capacitation Rate

The chlortetracycline (CTC) fluorescence assay was performed as previously described [10,19,20]. After thawing, spermatozoa were capacitated in 90 μL Cook’s medium for 0 h and 1 h (37 °C, 5% CO_2_). Hoechst was added 1:1000 and incubated for 20 min, then spermatozoa were centrifuged at 600× *g* for 5 min, the supernatant was removed and the spermatozoa were resuspended in Cook’s medium and mixed with an equal volume (30 μL/30 μL) of CTC solution (750 µM CTC, 20 mM Tris, 130 mM NaCl, and 5 mM cysteine, pH 7.8, all obtained from Sigma-Aldrich, Merck KGaA, Darmstadt, Germany). After 20 s, 10 μL of 4% paraformaldehyde in 0.5 M Tris buffer were added. After 5 min, spermatozoa were centrifuged at 600× *g* for 5 min, the supernatant was removed, and the spermatozoa were resuspended with 40 μL of PBS and 1% BSA. A volume of 10 μL of the sperm suspension was smeared onto a glass slide and covered with a cover slip. The samples were observed under a magnification of 100X with a Leica microscope with epifluorescence illumination. Spermatozoa with a bright fluorescence over the entire head were considered as uncapacitated (Pattern F), while those classified as capacitated showed a fluorescence-free band in the post-acrosomal region (capacitated with intact acrosome, Pattern B) or a uniform non-fluorescence pattern (capacitated without acrosome, that is, the spermatozoa underwent the acrosome reaction, Pattern AR) [20,21]. In our study, spermatozoa were considered as being capacitated when they were showing either pattern B or pattern AR. For each group, approximately 300 spermatozoa were assessed.

#### 2.7.4. Evaluation of the Acrosome Reaction

The acrosome reaction is defined as the percentage of spermatozoa without an acrosome at a defined timepoint compared to the total number of spermatozoa examined. The acrosome reaction was determined after incubation for 0 h and 1 h at 37 °C in 500 µL Cook’s medium (control) according to a modification of the Coomassie brilliant blue G-250 procedure [10,22]. After staining, spermatozoa were observed at a magnification of 40X using bright-field microscopy. When the acrosome was intact, the acrosomal ridge of the spermatozoa stained intensely blue, whereas acrosome-reacted spermatozoa were not stained. In total, 4 slides each with at least 200 spermatozoa were assessed to determine the acrosome reaction.

### 2.8. Statistical Analysis

The continuous variables are depicted as mean ± standard error of the mean (SEM). The Shapiro–Wilk test was used to check continuous data for normality. Differences between treatments according to age groups were assessed by two-sample *t*-test allowing for potential heterogeneity of the variances (method Satterthwaite). The fertilization rate (number of 2-cell embryos per oocyte) and the birth rate (number of pups per 2-cell embryo) were treated as binomially distributed variables. Likewise, capacitation and acrosome reaction were also assumed to be binomially distributed in the total sum of spermatozoa, yielding rates and proportions. Therefore, logistic regression was used to model the associations between the binomially distributed outcome variables and D-Asp treatment vs. control. The analyses were adjusted for age of the males (9 and 16 weeks) and duration of treatment (2 and 4 weeks). All possible interactions of the co-variables were considered, and insignificant main effects and insignificant interactions were removed by backward selection (option selection = backward in SAS procedure LOGISTIC). If in the logistic regression analyses the ‘Deviance Goodness-of-Fit Statistics (=deviance/degrees of freedom)’ was significantly greater than 1, correction for over-dispersion (option: scale = d) was employed. To be conservative in this respect, no correction was done in case of under-dispersion. In most cases, data complied well with the binomial assumption and no heterogeneity correction was necessary. Hormone levels in the testes were compared with *t*-test allowing for unequal variances. Two-by-two table chi-square tests were employed for the association between treatment and abnormalities (head, tail, cytoplasmic droplets, and total) by male age and duration of treatment. The values for concentration, total motility, progressive motility, and velocity distribution are descriptive in nature since samples from 2 straws were measured in duplicate. *p* values  ≤  0.05 were considered statistically significant. Statistical analyses were performed with the software package SPSS Statistics 23 (IBM Corp., Armonk, NY, USA) and with SAS/STAT software 9.4 (SAS Institute Inc.: SAS/STAT User’s Guide, Cary, NC, USA, 2014).

### 2.9. Ethical Review Procedure

All animal studies were approved by the Animal Care and Use Committee at CNR (Protocol number 0000079 of 18 January 2016). They were carried out in accordance with the guidelines approved by the Italian Ministry of Health in compliance with the Legislative Decree 26/2014 and performed according to the ARRIVE Guidelines [23].

## 3. Results

### 3.1. D-Asp Treatment Increases the In Vitro Fertilization (IVF) Rate

Males aged 5 or 7 weeks were administered an oral treatment of D-Asp for 4 or 2 weeks, respectively, so that they were 9 weeks old at the end of the treatment. The results of the IVF are shown in Table 1 and Figure 2a. After 2 and 4 weeks, the percentage of 2-cell embryos in the D-Asp-treated and control groups was 68.8% compared to 44.2%, respectively, (*p* = 0.0001) and 47.4% vs. 38.7% (*p* = 0.0189), respectively. In another experiment, males aged 12 or 14 weeks were treated with D-Asp for 4 or 2 weeks so that they were 16 weeks old at the end of the experiment. The results of the IVF are shown in Table 1 and Figure 2b. A 2-week treatment period significantly increased the IVF rate as compared to controls (54.7% vs. 29.6%; *p* = 0.0001). D-Asp administration also induced a significant increase of the IVF rate after 4 weeks of treatment compared to the controls (61.3% vs. 48.2%; *p* = 0.0011). Using the IVF rates in the control groups as basal values, in both 9-week-old and 16-week-old mice, a higher increase in the IVF rate was observed after 2 weeks of D-Asp treatment (55.7% and 84.8%, respectively) than after 4 weeks of D-Asp treatment (22.5% and 27.2%, respectively).

The IVF rate may show variations inherent to the environment and the males themselves. For this reason, all IVF results were compared by taking the D-Asp-treated sample and its corresponding control used in the same session into consideration and cannot be compared among different sessions.

### 3.2. The Birth Rate Is Not Affected by D-Asp Treatment

The birth rates are shown in Figure 3a,b. For all ages, the birth rates averaged 63.3–73.3% (controls) and 66.7–76.7% (D-Asp). No significant differences were observed between the D-Asp-treated and control groups.

### 3.3. Oral Administration of D-Asp Elevates Hormone Levels in Testes and in Sera

The results of the hormone measurements are shown in Figure 4. In testes, hormone levels were higher than those in sera. In 9-week-old treated males, the hormone levels were generally higher in both sera and testes than those in the controls at the respective timepoints. In testes, after 2 weeks of D-Asp administration, a significantly higher concentration of epitestosterone (11 ng/g vs. 4.1 ng/g; *p* = 0.0211) and testosterone (44 ng/g vs. 17 ng/g; *p* = 0.0006) were detected, but not of LH (370 ng/g vs. 361 ng/g). D-Asp treatment for 4 weeks significantly improved all hormone levels (LH: 2151 ng/g vs. 857 ng/g; *p* = 0.0001, epitestosterone: 27 ng/g vs. 17 ng/g; *p* = 0.0078, and testosterone: 27.3 ng/g vs. 14 ng/g; *p* = 0.0002). In addition, the hormone levels in pooled sera in the D-Asp-treated group were generally higher than those in the control group at both timepoints. With respect to 16-week-old males, after 2 weeks of D-Asp treatment, the epitestosterone and testosterone levels increased in testes compared to the controls (*p* > 0.05). The concentration of LH significantly increased in testes after 4 weeks of D-Asp administration (1063.7 ng/g vs. 598.8 ng/g; *p* = 0.0186) when epitestosterone and testosterone levels were also increased (*p* > 0.05). In addition, the hormone levels in pooled sera in the D-Asp-treated group increased 4 weeks after treatment. In the 16-week-old mice, the hormone levels were generally higher than those in the 9-week-old mice.

### 3.4. Total and Progressive Sperm Motility, but Not Concentration and Velocity Increases after D-Asp Treatment

The sperm concentration and motility were measured from two straws in duplicate. The sperm concentration in 9-week-old males as well as in 16-week-old males after 2 and 4 weeks of D-Asp treatment is shown in Table 2. The sperm concentration in 16-week-old males was higher than that for 9-week-old males. Notably, the number of spermatozoa is still in vast excess and allows a final sperm concentration within the limits established by the IVF protocol (2–6 × 10^5^/mL spermatozoa with a maximum of 100 oocytes per IVF dish).

In both age groups, total and progressive sperm motility were similar for D-Asp-treated and control groups after 2 weeks, while they were higher in the D-Asp-treated group after 4 weeks. In 9-week-old mice, 4 weeks of D-Asp treatment improved the total and progressive sperm motility compared to the controls (23.3% vs. 21.8% and 10.3% vs. 7.3%, respectively, Figure 5a,b). In 16-week-old mice, the total and progressive motility showed similar patterns; D-Asp administration for 4 weeks increased the total motility from 21% to 28.5% (Figure 5c) and the progressive motility from 7.8% to 10.3% (Figure 5d).

The velocity distribution of the spermatozoa was measured in duplicate and classified into rapid, medium, slow, and static, as presented in Figure 5e,f. For both age groups and for both D-Asp-treated and control groups within each age group, similar patterns were observed for each category of velocity. The percentage of static spermatozoa ranged from 71.3% (16-week-old mice after 4 weeks of treatment) to 85.3% (9-week-old controls at week 2). This was followed by the percentage of rapid (12%; 9-week-old control mice at 2 weeks, to 26%; 16-week-old mice after 4 weeks of treatment), medium (1.8%; 9-week-old mice of both control and treated group at week 4 and 16-week-old controls after 4 weeks, to 2.8%; 16-week-old mice after 2 weeks of treatment), and slow (0–0.5%) spermatozoa.

### 3.5. D-Asp Treatment for 2 Weeks Decreases Sperm Abnormalities in 9-Week-Old Males

Spermatozoa were examined morphologically and abnormalities of the head, the tail as well as the presence of cytoplasmic droplets were recorded. In 9-week-old males (Figure 6a), total abnormalities were significantly lower after 2 weeks of D-Asp treatment (14.3%) than in control spermatozoa (19.5%), *p* = 0.0482. No significant differences were observed 4 weeks after D-Asp treatment of 9-week-old males and in spermatozoa from 16-week-old males at both timepoints (Figure 6b).

### 3.6. The Capacitation Rate Is Increased after D-Asp Treatment

Spermatozoa were thawed and the capacitation status was measured immediately and after 1 h of incubation. The percentage of capacitated spermatozoa was expressed as the number of spermatozoa showing the B and AR patterns compared to total number of spermatozoa examined. The results obtained are depicted in Figure 7a–c. After treatment of 9-week-old males with D-Asp for 2 and 4 weeks (Figure 7a), compared to controls, the capacitation rate was significantly higher immediately after the thawing of the spermatozoa: 58.6% vs. 43.5%; increase by 34.7%; *p* < 0.0001 and 58.7% vs. 43.1%; increase by 36.2%; *p* < 0.0001, respectively, and at 1 h: (72.4% vs. 60.6%; increase by 19.5%; *p* = 0.0003 and 70.3% vs. 62%; increase by 13.4%; *p* = 0.0132), respectively. In 16-week-old males (Figure 7b), after 2 weeks of treatment, the capacitation rate significantly increased at thawing (81% vs. 62.8%; increase by 29%; *p* < 0.001) and after 1 h of incubation (88.6% vs. 73.7%; increase by 20.2%, *p* < 0.001), respectively. An increase in the capacitation rate by 9.6% (81.3% vs. 74.2%) was also observed in the 4-week-treated group after 1 h; *p* < 0.001 (Figure 7b). Summarizing, we observed that D-Asp-treatment has a significant impact on the capacitation rate of spermatozoa, which was more evident in samples from 9-week-old males since capacitation rates in D-Asp-treated 9-week-old males were similar to those of untreated 16-week-old males.

### 3.7. The Acrosome Reaction Increases after D-Asp Treatment

The acrosome reaction was evaluated immediately after thawing and after 1 h of incubation. The results obtained are shown in Figure 7d,e. In 9-week-old males (Figure 7d), D-Asp treatment significantly increased the acrosome reaction after 2 as well as after 4 weeks of treatment compared to controls (5.7% vs. 2.1%; increase by 171.4%; *p* < 0.05 and 6.2% vs. 2.0%; increase by 210%; *p* < 0.0001 after thawing and 38.4% vs. 32.7%; increase by 17.4%; *p* < 0.0001 and 46.3% vs. 32.4%; increase by 42.9%; *p* < 0.0001 after 1 h). For 16-week-old males (Figure 7e), the acrosome reaction was significantly higher in the 2 week-treated group (24.5% vs. 17.8%; increase by 37.6%; *p* < 0.0001 at thawing and 48.8% vs. 42.8%; increase by 14% after 1 h; *p* < 0.0001) and after 4 weeks of D-Asp treatment (23.8% vs. 19.1%; increase by 24.6%; *p* < 0.0001 at thawing and 50.2% vs. 42.5%; increase of 18.1% after 1 h; *p* < 0.0001). Generally, these results revealed a higher impact on spermatozoa from 9-week-old males compared to those from 16-week-old males. Similar to the capacitation rate, the impact of D-Asp was higher in 9-week-old males.

## 4. Discussion

We previously reported that treatment of 7-week-old male mice with 20 mM sodium D-Asp in drinking water led to increased sperm fertility after 2 and 4 weeks, but not after 6 weeks [14]. To determine if the effect of D-Asp was influenced by the age of mice at euthanasia, the duration of treatment, or both, we treated mice for 2 or 4 weeks so that they were 9 weeks (young) or 16 weeks old (adult) at the time of sperm cryopreservation. The present data demonstrate that oral administration of D-Asp increases sperm fertility in both young and adult male mice, regardless of the duration of treatment. Moreover, our results showed that in both young and adult mice, the highest increase in in vitro fertilizing capacity is obtained after 2 weeks of treatment (55.7% and 84.8%, respectively), confirming our previously published work [14]. This knowledge can be useful in cryorepositories, such as the European Mouse Mutant Archive (EMMA), where mice can be treated with D-Asp for a minimum of 2 weeks to obtain a significant improvement in sperm fertility. This approach can lead to a reduction in the number of oocyte donors for the production of embryos or zygotes for many purposes including genetic engineering, thereby saving time and resources. Furthermore, our data confirmed that embryo development is not affected by in vivo and in vitro D-Asp treatment, as previously reported [10,14].

D-Asp regulates the synthesis and secretion of several hormones in endocrine and neuroendocrine tissues [12]. Luteinizing hormone (LH) and testosterone are essential to initiate and maintain normal spermatogenesis [24]. Recently, Usiello et al. [25] showed that in rats, D-Asp stimulates the release of gonadotropin-releasing hormone from the hypothalamus, which induces the release of LH from the pituitary gland, ultimately resulting in increased testosterone biosynthesis. In addition, in both men and rats the hormone levels (LH and testosterone) significantly increased after oral administration of a dose of 20 mM D-Asp [26]. The present study showed that oral administration of sodium D-Asp in mice increases LH, epitestosterone, and testosterone levels in testes and in serum at both ages examined compared to controls. Moreover, we observed a higher concentration of hormone levels in the testis than in serum, which is in agreement with the literature [27]. We also noticed that the concentration of epitestosterone and testosterone is generally higher in adult males than in young males. Indeed, the testosterone level varies depending on the age [28,29] and strain [30,31] of mice. In contrast, the LH concentration in young and adult mice is not significantly different, as previously reported for rats [32]. Surprisingly, we also observed important variations in testosterone levels in the testes of 9-week-old males, which may be due to individual variation caused by episodic secretion of testosterone [28]. Finally, we found that after 2 weeks of D-Asp treatment, there is a strong increase in testosterone in the testes (2.6-fold in young males and 1.2-fold in adult males) while the LH concentration is stable (1.02-fold in both age groups). Although Leydig cells produce testosterone via LH stimulation, we observed that D-Asp could directly induce the synthesis of testosterone, as previously reported [12,33].

We observed that the enhanced IVF rate and hormone level elicited by D-Asp is not associated with an increase in sperm concentration in both young and adult males. These results are in contrast to those from two studies, which reported that 200 µM D-Asp for 30 min, 2 h, or 4 h stimulates proliferative pathways in GC-1 [34] and in GC-2 cells, two mouse spermatocyte-derived cell lines [35]. Furthermore, a negative effect of 10 mM, but not 1 or 5 mM D-Asp, added to the culture medium on mitosis in premeiotic germ cells was reported [13]. The reason for this discrepancy is not clear, but it might be due to varying effects on the different stages of spermatogenesis or different doses used for in vitro and in vivo experiments. While D-Asp can improve sperm fertility in vivo, doses that are used for in vitro work are much lower. This is exemplified by our previous publications [10,14]. For in vivo work, 20 mM was administered orally while 4 mM was used for in vitro work without negatively affecting the results

Sperm quality has a direct influence on the fertilization rate [36]. In our study, the improved fertilization capacity observed after D-Asp administration is associated with a general increase in total and progressive sperm motility. This confirms our previous results obtained using both in vivo and in vitro approaches [10,14]. Notably, there are other reports showing that D-Asp treatment led to an increase in sperm motility in rabbits [37], roosters [38], and men [39,40] in vivo, as well as in cattle [38,41] and men [40,42] in vitro.

Sperm morphology appears to be an important parameter to predict fertility in vivo [43]. We reported that D-Asp exerted a positive effect on spermatozoa from young males in vivo, reducing the percentage of morphological abnormalities, while in 16-week-old mice no significant differences were found after D-Asp treatment.

The present results showed that oral D-Asp administration significantly improved the capacitation rate and the acrosome reaction regardless of the duration of D-Asp treatment and the age of the males used. The capacitation process is a complex ensemble of several molecular events that lead to hyperactivated sperm motility, acrosomal exocytosis, and rapid fertilization requiring complex signaling cascades that include post-translational modifications [44,45]. It prepares the sperm to undergo the acrosome reaction with the accompanying release of lytic enzymes and exposure of membrane receptors, which are required for sperm penetration through the zona pellucida and for fusion with the oolema [46,47]. The assessment of the acrosome was shown to be a stable parameter of sperm function and a valid tool to predict the fertilizing potential of spermatozoa [48]. We recently showed that D-Asp had a positive impact on spermatozoa from young males in vitro by improving the capacitation rate and the acrosome reaction, thereby facilitating higher fertilization rates [10]. We therefore investigated whether D-Asp exerts the same effect in vivo. In the present study, as expected for untreated males, both the capacitation rate and the acrosome reactions were lower in young compared to adult males. In young males, both 2 and 4 weeks of D-Asp treatment increased the capacitation rate after thawing and after 1 h of incubation up to a level observed in adult control males. Similar to the capacitation rates, after D-Asp treatment, younger males showed an increase in the percentage of acrosome-reacted spermatozoa. The impact of D-Asp on the acrosome reaction in young males was higher than that in adult males, where we also observed a significant improvement. In our previous work in vitro [10], we only detected a positive effect of D-Asp in younger (9- and 11-week-old) males, but not in older (13- and 16-week-old) males. This result may be due to the different effects of in vitro and in vivo treatment.

The mechanisms by which D-Asp treatment increases sperm capacitation and acrosome reaction are not well understood. Spermatozoa which were capacitated in vitro show hyperpolarization of the membrane potential and regulates zona pellucida-dependent acrosomal secretion, which is necessary for the acrosome reaction in mice [49]. A key capacitation marker is the activation of a specific signal transduction pathway leading to protein tyrosine phosphorylation [50,51]. It is driven by cyclic adenosine monophosphate (cAMP) and is modulated by the redox status of the cells [52,53,54,55,56]. Interestingly, an elevation of the cAMP synthesis in rat Leydig cells after in vitro treatment with D-Asp was reported [26]. Furthermore, oxidative processes may play a role. Indeed, even though oxidative stress is known to induce decreased fertilizing potential and alter sperm motility, a low level of reactive oxygen species is still important for mechanisms influencing fertility [57]. Notably, it stimulates tyrosine phosphorylation, facilitates cholesterol efflux, and enhances cAMP generation, all of which are necessary cellular processes implicated in the regulation of sperm capacitation and acrosome reaction [58]. On the other hand, there are also some reports on the antioxidant effect of D-Asp [59,60]. It would be of interest to investigate the pro-oxidant environment generated by the D-Asp treatment and its potential influence on sperm quality and fertility in future studies.

Further studies are necessary to continue unveiling the mechanistic action of D-Asp, but it is clear that this molecule plays an important role in reproduction and the fertilizing capacity of mammalian spermatozoa. In future studies, we will also investigate if D-Asp treatment improves IVF in infertile or subfertile genetically engineered mice.

## 5. Conclusions

We demonstrated that D-Asp administration in vivo increases fertility, hormone levels, and sperm quality already after 2 weeks in both young and adult male mice. As observed in vitro [10], we showed that D-Asp has a direct effect on sperm capacitation and the acrosome reaction in vivo. Here, we extended the previous in vivo study [14] reporting the positive effect of D-Asp not only in young, but also in adult, animals. Interestingly, the spermatozoa concentration did not change after D-Asp treatment, indicating a role of D-Asp in sperm maturation rather than in spermatogenesis for the study period. Notably, after D-Asp treatment, the capacitation rate and the acrosome reaction from spermatozoa from young males reached levels comparable to those of untreated adult males.

## Figures and Tables

**Figure 1 animals-12-01350-f001:**
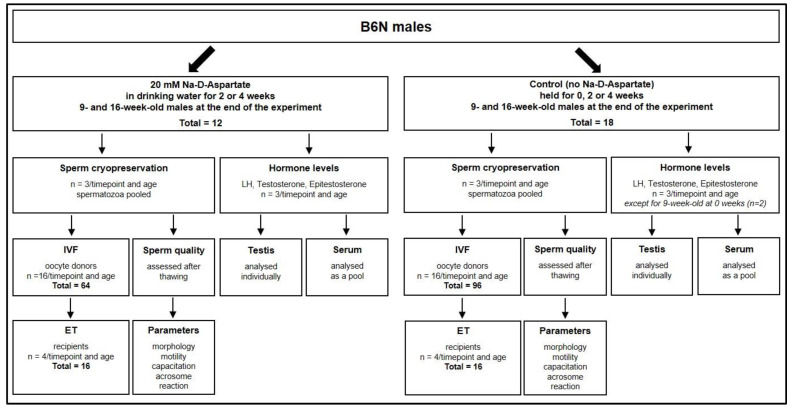
Experimental design of the study. LH: Luteinizing Hormone. IVF: In Vitro Fertilization. ET: Embryo Transfer.

**Figure 2 animals-12-01350-f002:**
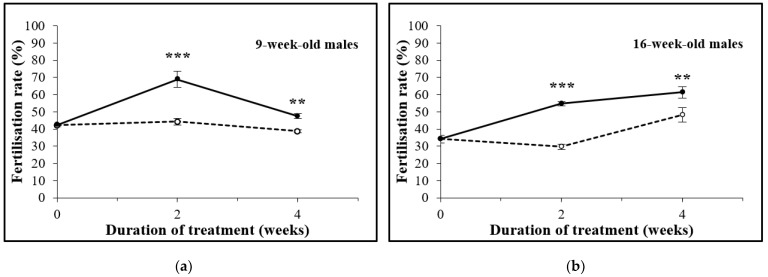
The in vitro fertilization rate with spermatozoa obtained from 9-week-old (**a**) and 16-week-old males (**b**) treated (continuous lines) or not treated (dotted lines) with D-Asp. Significant differences between D-Asp-treated males and controls are marked as follows: **: *p* < 0.01, ***: *p* < 0.001.

**Figure 3 animals-12-01350-f003:**
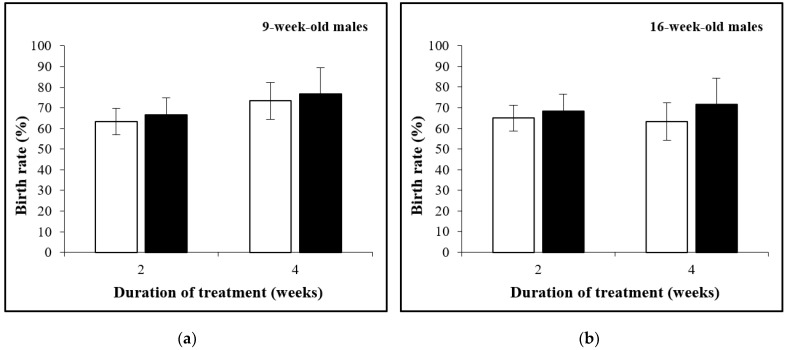
The birth rate with spermatozoa obtained from 9-week-old (**a**) and 16-week-old males (**b**) treated (black) or not treated (white) with D-Asp.

**Figure 4 animals-12-01350-f004:**
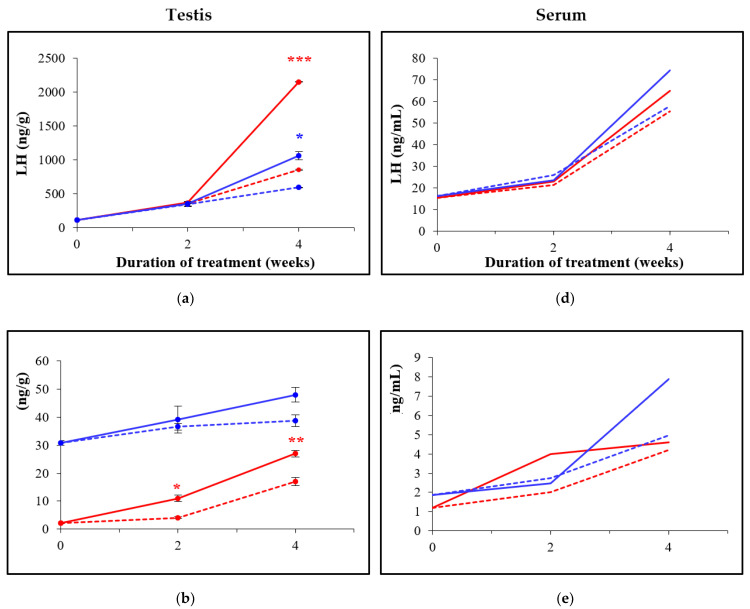
Hormone measurement in males treated with D-Asp (continuous lines) and control males (dotted lines) from 9-week-old (red lines) and 16-week-old males (blue lines) in testes (**a**–**c**) and in sera (**d**–**f**). With respect to the hormone levels in the testis, the results are the mean of three samples (*n* = 3) and are expressed as mean ± SEM. The hormone levels in serum were measured in triplicates using three samples that were previously pooled. Significant differences between D-Asp-treated and control 9-week-old males are marked with red asterisks as follows: *: *p* < 0.05, **: *p* < 0.01, ***: *p* < 0.001, whereas in D-Asp-treated and control 16-week-old males, this is marked with a blue asterisk as follows: *: *p* < 0.05.

**Figure 5 animals-12-01350-f005:**
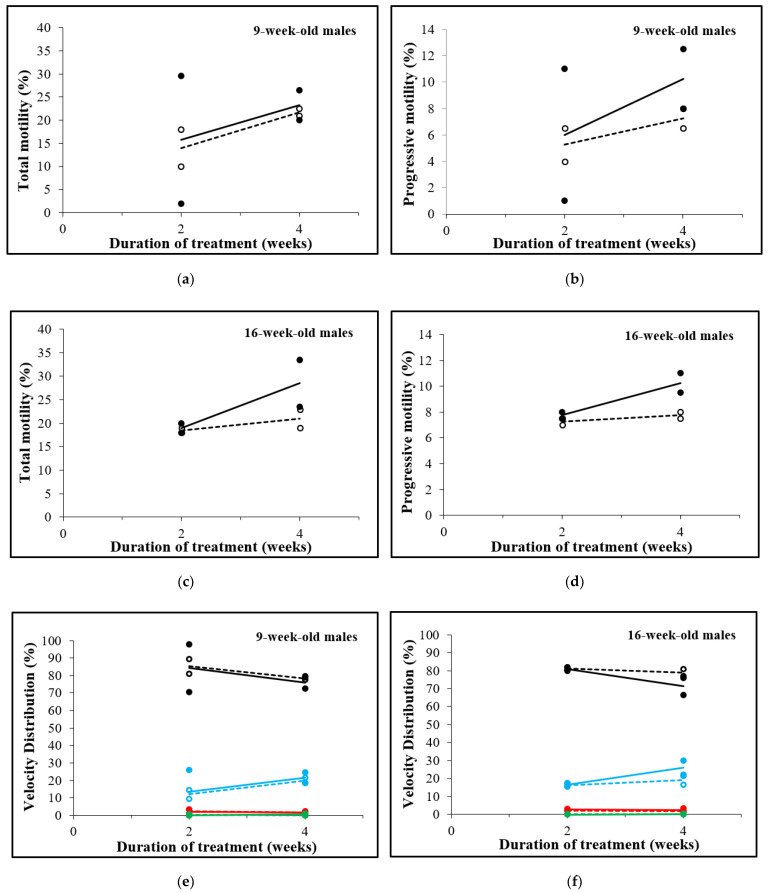
Total motility, progressive motility, and velocity distribution (rapid: blue lines; medium: red lines; slow: green lines; and static: black lines) of spermatozoa from 9-week-old (**a**,**b**,**e**) and 16-week-old males (**c**,**d**,**f**) treated (continuous lines, single values are shown in solid circles) or not treated (dotted lines, single values are shown in open circles) with D-Asp. The results are shown as the mean of two straws measured in duplicate each.

**Figure 6 animals-12-01350-f006:**
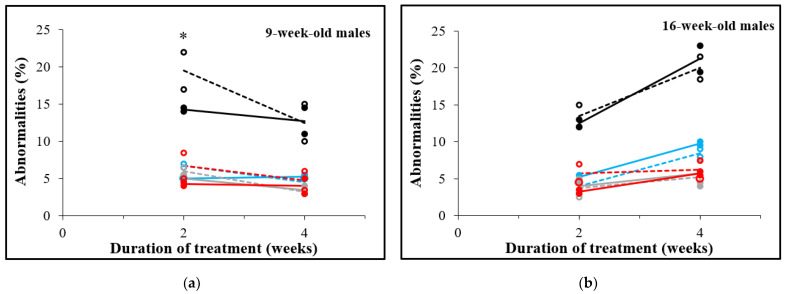
Abnormalities (total: black lines; head: blue lines; tail: gray lines; and cytoplasmic droplet: red lines) of spermatozoa from males treated (continuous lines, single values are shown in solid circles) or not treated (dotted lines, single values are shown in open circles) with D-Asp. Males were either 9 weeks old (**a**) or 16 weeks old (**b**). The results are shown as a mean of two replicates of 200 spermatozoa each. Significant differences between D-Asp-treated males and controls are marked as follows: *: *p* < 0.05.

**Figure 7 animals-12-01350-f007:**
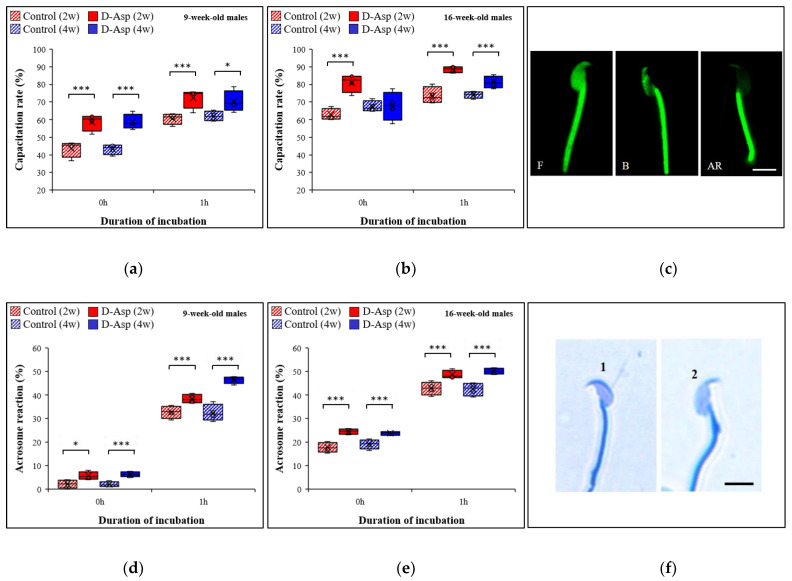
The capacitation rate of spermatozoa from control males and from males treated with D-Asp for 2 weeks or 4 weeks, which were either 9 weeks old (**a**) or 16 weeks old (**b**) at 0 h and 1 h. (**c**) Representative image of CTC staining: non-capacitated (F), capacitated (B), and acrosome-reacted (AR) spermatozoa. Scale bar = 3 µm. The acrosome reaction of spermatozoa from control males and from males treated with D-Asp for 2 weeks or 4 weeks at 0 h and 1 h. Males were either 9 weeks old (**d**) or 16 weeks old (**e**). (**f**) Representative image of Coomassie brilliant blue G-250 staining. (1) spermatozoa with acrosome, (2) spermatozoa without acrosome. Scale bar = 3 µm. Box-and-whiskers plot; data points: open circles, maximum: endpoint of upper whisker, minimum: endpoint of lower whisker, third quartile (75th percentile): upper edge of the box, first quartile (25th percentile): lower edge of the box, median (50th percentile): line inside the box, mean: X. Significant higher rates in D-Asp-treated males than in controls: *: *p* < 0.05, ***: *p* < 0.001.

**Table 1 animals-12-01350-t001:** The number of oocytes used and 2-cell embryos obtained with spermatozoa from 9-week-old and 16-week-old males.

	9-Week-Old Males				16-Week-Old Males			
	Control		D-Asp Treatment		Control		D-Asp Treatment	
Duration of Treatment (Weeks)	No. of Oocytes	No. of 2-Cell Embryos	No. of Oocytes	No. of 2-Cell Embryos	No. of Oocytes	No. of 2-Cell Embryos	No. of Oocytes	No. of 2-Cell Embryos
0	283	120	n.d.	n.d.	318	108	n.d.	n.d.
2	242	108	260	175	316	93	273	149
4	346	134	349	166	333	158	322	194

n.d.: not done.

**Table 2 animals-12-01350-t002:** Sperm concentration (million/mL) of 9-week-old and 16-week-old males (mean ± SD).

Duration of Treatment (Weeks)	9-Week-Old Males		16-Week-Old Males	
	Control	D-Asp	Control	D-Asp
2	13.8 ± 1.3	13.0 ± 4.7	30.8 ± 0.2	22.4 ± 0
4	7.4 ± 1.1	9.1 ± 1.5	25.5 ± 3	21.7 ± 5.7

## Data Availability

All data generated or analyzed during this study are included in this published article.

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
