# Peer review of "Oral D-Aspartate Treatment Improves Sperm Fertility in Both Young and Adult B6N Mice"

_animals, 2022, doi:10.3390/ani12111350_

Round 1
Reviewer 1 Report
Line 232: 2.9. Ethical Review Procedure
Superovulation and embryo transfer are experimental procedures which require a license. In this chapter no animal experiment license number is mentioned.
Author Response
Reviewer 1
- Line 232: 2.9. Ethical Review Procedure: Superovulation and embryo transfer are experimental procedures which require a license. In this chapter no animal experiment license number is mentioned.
In accordance with the Ministry of Health, procedures and archiving/freezing operations including superovulation, embryo transfer and euthanasia are considered zootechnical procedures and authorized internally (Animal Welfare Body and Ethical Committee). Standard Operating Procedures (SOPs) were authorized internally by the CNR-EMMA/Infrafrontier Animal Welfare Body (Organismo Preposto al Benessere degli Animali, OPBA) of the Institute of Biochemistry and Cell Biology (Protocol number 0000079 of 18 January, 2016). We added the protocol number in the manuscript: Lines 238, 521-524.

Reviewer 2 Report
The authors extended their research on the role of D-Aspartate in the maturation of spermatozoa in vivo, including those cryopreserved, and particularly in adult mice where oral treatment improves IVF rates.
Spermatozoa were screened and a positive effect was detected for several fertility parameters like motility, morphology, capacitation and acrosome reaction.
The authors showed that D-Asp treatment increases hormone levels related to male reproductive function. Indeed in this work, authors show that a short period of oral treatment can have an impact, not only on young (as previously reported) but also on adult B6N male mice.
Although the mechanism of action of D-Asp is still unclear, this work supports the hypothesis that it plays a role in sperm maturation rather than on spermatogenesis at least for the period analysed. Authors found that treatment does not affect sperm concentration nevertheless all fertility parameters were improved.
These findings will be helpful for cryo-repositories and transgenic units to save resources and maintain mice more efficiently, reducing the number of oocyte-donor females used in IVF procedure therefore fully respecting the 3R principles.
Such a simple, time-saving and convenient method to improve male fertility could be very important in the field of animal research.
The title indicates the aim of the manuscript and the abstract clearly indicates the work objective, methodology and result of the study. The methodology is well articulated and the description is well made.
Manuscript is well-structured and scientifically sound therefore, The conclusions are consistent with the evidence and arguments presented.
I have some suggestions:
Line 233: Given that in this procedure the animals were euthanized (Line 134), Ministry of Health approval / authorization number is missing. Please add it.
Line 235: Remove the legislative decree 116/92 because is no longer into force.
I support the publication of this manuscript after the abovesaid minor revision.
Author Response
Reviewer 2
- Given that in this procedure the animals were euthanized (Line 134), Ministry of Health approval / authorization number is missing. Please add it.
In accordance with the Ministry of Health, procedures and archiving/freezing operations including superovulation, embryo transfer and euthanasia are considered zootechnical procedures and authorized internally (Animal Welfare Body and Ethical Committee). Standard Operating Procedures (SOPs) were authorized internally by the CNR-EMMA/Infrafrontier Animal Welfare Body (Organismo Preposto al Benessere degli Animali, OPBA) of the Institute of Biochemistry and Cell Biology (Protocol number 0000079 of 18 January, 2016). We added the protocol number in the manuscript: Lines 238, 521-524.
- Remove the legislative decree 116/92 because is no longer into force.
As suggested by this reviewer, we removed mention of the information concerning the legislative decree 116/92.
Reviewer 3 Report
In this manuscript, Raspa et al. investigated the effect of D-Asp on sperm function and found that D-Asp could improve several sperm function especially in vitro fertilization ratio after sperm cryopreservation. The manuscript was written well. However, much data is not explained well and needs to be improved. Please find the comments below.
Line 32, it is not clear what the sperm quality means. Please specify, instead.
In the introduction, the rationale of using cryopreserved sperm is not really up to standard. Please expand a better explanation for this in the introduction.
Figure 2 legend, (__) is a little confusing and unusual. Please replace with (—) or use text directly (solid line/dash line).
Is there a reason that authors tested the hormone levels from pooled sera? That is not comparable with other tests and might not reflect in a good way.
As shown in Figure 4, the hormone level in group of D-Asp is increased. It has been discussed from line 409, hormones are necessary for spermatogenesis. But in table 2, the sperm concentration in 16-week-old mice is lower than that in control. What is the underlying mechanism for this controversy?
There are quite some figures not showing any n value (figure4-6, table 2). Are they one time trial? If so, it is not sufficient to make any conclusion. If not, please indicate mean value and SEM/SD. Please specify.
Figure 7, it is very unusual the capacitation ratio in C57 mice is so high even with 0h capacitation (40-60%)? What is the reason for this?
Author Response
Reviewer 3
- Line 32, it is not clear what the sperm quality means. Please specify, instead.
As suggested by the reviewer, the sperm quality was specified in the abstract (Line 29) and the introduction (Line 69-70). This refers to morphology, abnormalities, motility and velocity.
- In the introduction, the rationale of using cryopreserved sperm is not really up to standard. Please expand a better explanation for this in the introduction.
We explained the rationale for using cryopreserved sperm in the introduction. We extended the previous information by adding on lines 64-66 “Cryopreserved spermatozoa were used to simulate routine work and since use of fresh spermatozoa would make multiple analyses on the same sample extremely difficult.”
- Figure 2 legend, (__) is a little confusing and unusual. Please replace with (—) or use text directly (solid line/dash line).
As suggested, the figure legend was modified.
- Is there a reason that authors tested the hormone levels from pooled sera? That is not comparable with other tests and might not reflect in a good way.
We used pooled sera to compare our results with our previously published work, where we examined pooled samples. In addition, the pools allowed us to reduce the number of analyses limiting experimental costs.
- As shown in Figure 4, the hormone level in group of D-Asp is increased. It has been discussed from line 409, hormones are necessary for spermatogenesis. But in table 2, the sperm concentration in 16-week-old mice is lower than that in control. What is the underlying mechanism for this controversy?
Table 2 shows that after D-Asp treatment sperm concentration in 16-week-old mice is lower than that in the controls. However, this difference is within the normal individual variability, that is, 13-45 million/mL for B6, as previously reported by Sztein et al. (Jorge M. Sztein, Jane S. Farley, Larry E. Mobraaten, In Vitro Fertilization with Cryopreserved Inbred Mouse Sperm, Biology of Reproduction, Volume 63, Issue 6, December 2000, https://doi.org/10.1095/biolreprod63.6.1774) and is in line with our experience. Furthermore, this observation confirms that D-Aspartate does not stimulate spermatogenesis for a short treatment period of 2 to 4 weeks, as administered in the present study. Notably, the duration of spermatogenesis is approximately 34.5 days (Oakberg EF. Duration of spermatogenesis in the mouse and timing of stages of the cycle of the seminiferous epithelium. Am J Anat. 1956;99:507-16) so that an effect of D-Asp on sperm concentration may only possibly be observed after this time period. The duration of treatment in the present study would only allow effects on sperm maturation to be observed, as reported.
- There are quite some figures not showing any n value (figure 4-6, table 2). Are they one time trial? If so, it is not sufficient to make any conclusion. If not, please indicate mean value and SEM/SD. Please specify
More information was previously given in the respective sections in materials and methods and in the statistical section. We have specified the n values for these figures: Figure 4: Lines 299-301, 134-142, Figure 5: Lines 149-152, 159-160, 337-338, Figure 6: Lines 149-152, 350-351, Table 2: Lines 149-152, 307.
- Figure 7, it is very unusual the capacitation ratio in C57 mice is so high even with 0h capacitation (40-60%)? What is the reason for this?
Capacitation refers to the maturation status of the spermatozoa and their ability to penetrate and fertilize an oocyte. In previous and in the current studies, we used a staining that reveals the absence of the acrosome. Some of the spermatozoa would have already reacted and some react (capacitate) during incubation. Therefore, our results consider all capacitated spermatozoa (B + AR, as described in material and methods). Furthermore, the experimental procedure requires a period of incubation in Hoechst for 20 min before CTC staining, which could also increase the capacitation rate (timepoint 0, immediately after thawing). Notably, we have made similar observations in previous studies.

Round 2
Reviewer 3 Report
The reviewer thank authors for taking my comments. The manuscript has been revised well. Following up the previous question 4-6, there are quite some figures are from just 2 biological repeats which are usually not sufficient to make a conclusion. Please show both mean and SD/SEM when the assay was done with more than 1 sample. Hormone detection from pooled sera for saving cost also reduce the reliability and significance of this study.
Author Response
Reviewer 3
The reviewer thank authors for taking my comments. The manuscript has been revised well. Following up the previous question 4-6, there are quite some figures are from just 2 biological repeats which are usually not sufficient to make a conclusion. Please show both mean and SD/SEM when the assay was done with more than 1 sample.
We thank the reviewer for this constructive suggestion.
For routine conditions, usually spermatozoa from 2 males are cryopreserved. As shown in the experimental design in Figure 1, spermatozoa from 3 males per timepoint and group were pooled and straws from this pool were used for all corresponding experiments: concentration, total motility, progressive motility, velocity distribution, abnormalities, capacitation rate, and acrosome reaction. In total, 12 males were treated with D-Asp and 18 were used as controls. The figures where 2 straws were used are Figure 5 (Total motility, progressive motility, and velocity distribution) and Figure 6 (Abnormalities).
Minor editing was done to the statistical analysis: Line 227, 231, 307, 314.
Table 2: The mean and SD are now presented.
Because several strata (treatments, animal age, timepoints etc.) are combined in a larger overall data set and evaluated in ordinary or generalized ANOVA frameworks the present data are considered meaningful.
Figure 5 (Total motility, progressive motility, and velocity distribution)
The graphs were edited to include single values instead of presenting mean/SD. Lines 338-339: The legend was extended to include …” treated (continuous lines, single values are shown in solid circles) or not treated (dotted lines, single values are shown in open circles) with D-Asp”….
We used an ANOVA scheme to perform an orienting power analysis. Stable SDs were estimated by pooling all SDs in case single SDs were too variable. This is consistent with the assumption of variance homogeneity inherent in ordinary ANOVA. With very small sample sizes of n=2 or n=3 SDs may vary considerably by mere chance. This can be avoided by computing SDs across several strata.
Without loss of generality, one can always divide the mean by the SD (=null effect in the control group) and calculate the power for an effect 1.5 or 2.0 times as large as this null effect. I have attached the result of these analyses below for the two age groups investigated. In most cases, a sufficient power of > 0.80 was already achieved using n = 2 or n = 3. The following table compiles power values for ANOVA based on standardized group means (means/SDs) and a multiplicative global hypothetical effect of 1.5 for group sizes n=2, 3, and 4.
Figure 6 (Abnormalities)
The graphs were edited to include single values instead of presenting mean/SD. Lines 351-352: The legend was extended to include …” treated (continuous lines, single values are shown in solid circles) or not treated (dotted lines, single values are shown in open circles) with D-Asp”….
We used an ANOVA scheme to perform an orienting power analysis. Without loss of generality, one can always divide the mean by the SD (=null effect in the control group) and calculate the power, e.g., for an effect 1.5 or 2.0 times as large as this null effect. I have attached the result of these analyses below for the two age groups investigated. In most cases, a sufficient power of > 0.80 was already achieved using n = 2 or n = 3.
Hormone detection from pooled sera for saving cost also reduce the reliability and significance of this study.
As shown in Fig. 4 and although sera were pooled, the data for both single testes (with some significance found) and sera showed generally higher hormone concentrations in the D-Asp group compared to the controls. Notably, these data also support previous reports (Raspa et al. 2018) where testes (n = 4) and sera (n = 4) were investigated singly.
